# Perinatal depression and adverse child growth outcomes in low-income and middle-income countries (LMICs): A systematic review and meta-analysis

Elizabeth Carosella[1], Shradha Chhabria[1,¤a], Hyelee Kim[1,¤b]*, Aliya Moreira[1,¤c], Dana Naamani[1,¤d], Brennan Ninesling[1,¤e], Aimee Lansdale[2‡], Lakshmi Gopalakrishnan[2‡], Bizu Gelaye[1], Aisha Yousafzai[1], Stefania Papatheodorou[1,3]

1 Harvard T.H. Chan School of Public Health, Boston, Massachusetts, United States of America, 2 University of California San Francisco, San Francisco, California, United States of America, 3 Rutgers School of Public Health, Piscataway, New Jersey, United States of America

☯ These authors contributed equally to this work.
¤a Current address: University Hospitals Cleveland Medical Center and Rainbow Babies and Children's Hospital, Cleveland, Ohio, United States of America
¤b Current address: University of California San Francisco, San Francisco, California, United States of America
¤c Current address: Beth Israel Deaconess Medical Center, Boston, Massachusetts, United States of America
¤d Current address: Duke University Medical Center, Durham, North Carolina, United States of America
¤e Current address: University of California San Diego, California, United States of America
‡ AL and LG also contributed equally to this work.
* Hyelee.Kim@ucsf.edu

**Data Availability Statement:** We added the data file (CSV format) as supporting information.

## Abstract

Perinatal depression (PND), which encompasses the antepartum and postpartum depression (APD and PPD), is a neglected crisis in low-income and middle-income countries (LMICs). We aimed to systematically search and meta-analyze existing evidence to determine whether a mother's PND affects adverse growth outcomes in children in LMICs (PROSPERO protocol: CRD42021246803). We conducted searches, including nine databases (PubMed, EMBASE, Web of Science, CINAHL Plus, Global Health Database, Google Scholar, WHO Regional Databases, PsycINFO, and LILACS) from January 2000 to September 2023. We restricted studies that assessed PND using validated screening tools or clinical interviews during pregnancy or within 12 months postpartum. We included studies that reported four types of adverse child growth outcomes (stunting, wasting, underweight, and overweight/obesity) in children younger than 5 years. We assessed the quality using the Newcastle Ottawa Scale and pooled risk ratios (RRs) and odds ratios (ORs) between PND and each adverse growth outcome using random-effects models. In total, 27 studies met the inclusion criteria for systematic review, with 24 eligible for meta-analysis, spanning data from 15 countries and 26,261 mother-baby pairs. Based on the studies that reported ORs, children below the age of 3 years with mothers experiencing PND had higher odds for stunting (OR 1.63, 95% CI 1.32, 2.02, $I^2$ = 56.0%) and underweight (OR 2.65, 95% CI 1.90, 3.68, $I^2$ = 34.5%) compared to children of mothers without PND. The pooled RRs for stunting

**Funding:** The authors received no specific funding for this work.

**Competing interests:** The authors have declared that no competing interests exist.

and underweight did not show significant differences between mothers with and without PND. Studies on wasting (n = 5) and overweight/obesity (n = 2) were limited, demonstrating inconsistent results across studies. The association between PND and adverse growth outcomes varied according to the measure of association, region, country, PND type, outcome timepoint, and study design. There were limited studies in diverse LMICs, particularly on wasting, or overweight/obesity as an outcome.

## Introduction

Perinatal depression (PND), a nonpsychotic depressive episode of mild to major severity, affects 11.9% of mothers during pregnancy or after delivery worldwide [1], with significant reductions in quality of life and day-to-day functioning, with potentially far-reaching consequences for ranges of adverse child health outcomes [2, 3]. PND encompasses depression among mothers during pregnancy, antepartum depression (APD), and depression after delivery, postpartum depression (PPD), depending on the timing of symptom onset [4, 5]. Research has suggested a higher prevalence of PND in low-income and middle-income countries (LMICs) compared to high-income countries where epidemiologic estimates are well studied [1, 6], and in LMICs, over 25% and 19% of women suffer from PND and PPD, respectively [7].

Maternal health during and after pregnancy plays a critical role in children's health. The first 1,000 days from conception to two years of age is a particularly sensitive period for children's development and sets the stage for their future growth trajectories. Moreover, eating behaviors and food preferences, influenced by parental care practices, become established during the first five years [8, 9]. A systematic review and meta-analysis in Africa highlighted a strong association between APD and adverse birth outcomes, such as preterm birth and low birth weight, while PPD was also highly associated with infant growth and health outcomes [10]. Further, studies have shown that PPD in LMICs could contribute to poor infant attachment and interactions with caregivers, potentially contributing to developmental and emotional challenges in children's future [6]. Inadequate nutrition during this time may lead to negative health outcomes for children, including cognitive impairment, compromised immune systems, and an increased risk of cardiometabolic disease [11–13].

Globally, nearly half of child deaths are associated with malnutrition [14]. According to the 2021 UNICEF/WHO/World Bank Group Joint Child Malnutrition Estimates, 149.2 million children below five years were affected by stunting, 45.4 million by wasting, and 28.9 million by overweight [15]. Despite progress in recent decades, stunting and wasting rates remain above-established targets, while the prevalence of overweight is on the rise [16]. Neglecting to address PND may be contributing to these trends, with severe consequences for children's health, development, and future economic participation [17]. Moreover, the COVID-19 pandemic is expected to heighten the risk of all forms of child malnutrition and poor maternal mental health [18, 19]. As part of efforts to achieve Sustainable Development Goal 2 and end all forms of malnutrition by 2030 [20], maternal mental health merits higher priority on the global public health agenda. Nonetheless, there is relatively less evidence regarding the relationship between perinatal maternal mental health and child adverse growth outcomes in LMICs [21, 22]. A synthesis of the available evidence for this relationship is useful for understanding these relationships and for highlighting gaps in the existing literature.

The objective of this systematic review and meta-analysis was to assess whether PND has an impact on four types of adverse growth outcomes (stunting, wasting, underweight,

overweight/obesity) among children under five years in LMICs using existing studies. While we considered the associations reported in the previous studies, our primary interest lies in understanding the temporal relationship between PND and adverse child growth outcomes. Thus, we hypothesized that the PND would exhibit a statistically significant association with adverse child growth outcomes in the pooled analyses. In contrast to previous meta-analyses on this topic, we intended to estimate the effect of exposures to APD and PPD separately or as a composite PND on a broader range of adverse growth outcomes, including wasting and overweight/obesity.

## Methods

We defined PND as a diagnosis made during pregnancy or within 12 months postpartum. Our primary exposure of interest was PND. Therefore, in cases where the authors reported maternal depression instead of PND, we limited the measurement time point for PPD to a period of up to 12 months after delivery to prevent reverse causation between the exposure, PPD, and the outcome. We reported the pooled odds ratios (ORs) and the pooled risk ratios (RRs) for child adverse growth outcomes for both disaggregated exposures (APD and PPD) and the composite exposure (PND) [4]. Given that each growth outcome has unique environmental and biological underpinnings, our aim was to report multiple disaggregated adverse growth outcomes, specifically stunting, underweight, wasting, and overweight/obesity. This study adhered to the pre-registered protocol on PROSPERO International prospective register of systematic reviews (registration number: CRD42021246803) [23].

Stunting was defined as height or length-for-age z-scores (HAZ or LAZ) of two standard deviations (SD) or more below the WHO Child Growth Standards median by sex [24]. Underweight was defined as weight-for-age z-scores (WAZ) of two SDs or more below the WHO Child Growth Standards median by sex [24]. Wasting was defined as weight-for-length or height-for-length z-scores (WLZ or HLZ) of two SDs or more below the WHO Child Growth Standards median by sex [24]. Overweight was defined as WLZ or HLZ of two SDs or more from the WHO Child Growth Standards median by sex, and obesity was defined as WLZ or HLZ of three SDs or more from the WHO Child Growth Standards median by sex [24]. Given the nature of the systematic review, the institutional review board approval was not required.

### Data source

We conducted searches in the following databases: PubMed, EMBASE, Web of Science, CINAHL Plus, Global Health Database (EBSCO), Google Scholar, WHO Regional Databases (Global Index Medicus), PsycINFO, and Latin American and Caribbean Health Sciences Literature (LILACS). Our initial search was from January 2000 to March 2021, and during the revision process, we updated the search until September 2023 to include the recent publications for the past two years. The exact search terms used for each of the databases are provided in S1 Text. Additionally, we supplemented our search with a hand search of the Women's Mental Health Journal, Conference abstracts of the International Marcé Society for Perinatal Mental Health, the WHO working group on PND, and related information and studies from websites of the Center for Global Mental Health and ReliefWeb.

### Study selection and data extraction

Identified references were uploaded to Covidence.org for the review process [25]. Eight reviewers completed title and abstract screening, full-text review, and data extraction (E.C., H. K., B.N., S.C., D.N., A.M., A.L., and L.G.). Each title and abstract, and subsequently each full-text study, underwent reviewed by a combination of two authors among the list (E.C., H.K., B.

N., S.C., D.N., A.M., A.L., and L.G.). Articles agreed by two authors were included in the study. For articles selected by only one author, the inclusion was based on the team's discussion of its suitability for inclusion.

Studies were considered eligible if they met the following criteria: 1) was an original research article published in a peer-reviewed journal; 2) contained demographic and clinical information on mother-baby pairs; 3) the exposure was either APD or PPD, diagnosed during pregnancy or up to 12 months after delivery; 4) depression diagnosis was based on a validated patient questionnaire or clinical interview; 5) the outcome was classified as stunting, wasting, underweight, overweight, or obesity among children under age five; 6) conducted in an LMIC as defined by the World Bank in 2021, including 136 low-, lower-middle, or upper-middle income countries [26]; 7) reported relevant quantitative measures, such as ORs, RRs, or other metrics suitable for meta-analysis (e.g., % proportion or beta-coefficients); and 8) the publication was available in English, Spanish, Portuguese, or French.

We excluded studies that were literature reviews, systematic reviews and/or meta-analyses, and clinical trials. When a study conducted secondary data analysis on data from a randomized controlled trial (RCT) as an observational study, we assumed that the path from randomization to PND did not exist due to the inclusion of randomization. Consequently, observational studies conducted within RCTs were included if there was no backdoor path between PND and adverse growth outcomes. If the studies used PND measurement addressing heterogeneous symptoms of depression, anxiety, and psychological distress, we included them if the association for depression was reported or if the authors demonstrated that the measure was validated for depression in the study population. In cases where adverse growth outcomes were defined outside the inclusion criteria, we included the studies based on consensus. Studies were excluded if they solely assessed adverse birth outcomes (e.g. low birth weight) or if the definition of adverse growth outcomes was omitted in the paper.

Since our research question focuses on adverse growth outcomes, we excluded studies that analyzed the continuous growth outcome indicators, such as height-/length-for-age Z score (HAZ/LAZ), weight-for-age Z score (WAZ), or height-/length-for-weight Z score (HWZ/LWZ), as opposed to adverse growth outcomes (binary variable). Additionally, during the full-text review, it became apparent that many studies examined the association between maternal depression outside the PND timeframe (e.g. maternal depression measured four years after delivery) and child growth outcomes that were not classified as adverse. While these findings were not the primary focus of our research, we have included a summary in the S1 Table due to their relevance to our main research question. This table includes 30 studies, which including additional findings from four studies within the final selection for systematic review and meta-analysis [27–56].

Standardized data collection forms were developed by two authors (B.N. and E.C.), and data extraction was performed by a combination of two authors among eight authors (D.N., H.K., S.C., A.M., B.N., E.C., A.L., and L.G) during May 2021 for the first search and October 2023 for the search update. The data extracted included the following variables: authors, publication year, country, study design, number of mother-child pairs, PND measurements, and their cut-points and timing, the type and definition of adverse growth outcomes, including the outcome timepoint, covariates, measures of association (types and values), and 95% confidence intervals (CIs).

## Quality assessment

The quality of the included studies was assessed by five authors (B.N., D.N., H.K., A.L., and L. G.), who evaluated studies using the Newcastle Ottawa Scale (NOS) (S2 Table) [57]. Each

study underwent assessment by two reviewers (D.N. and H.K.), and a third reviewer (B.N., A. L., or L.G.) provided the final approval and acted as a tiebreaker in the case of any discordance. One component of quality assessment for cohort studies examined whether the outcome of interest was absent at the baseline. However, given our research focus on the path from APD through adverse birth outcomes, we decided that this requirement was irrelevant to our research question. Consequently, we did not deduct any points for this component, regardless of whether the outcome was reported at baseline.

Studies with a score of 7 or higher out of a total score of 9 were categorized as "good quality." Nearly all studies, except for Joshi and Raut (2019), got scores of seven or above on the NOS scale, demonstrating sufficient quality for inclusion in the study. It is worth noting that while these studies were deemed to have good quality, each study may not have been entirely free from bias due to various factors, such as missing observations, residual and unmeasured confounding, and measurement bias. Joshi and Raut (2019) received a score of 5 due to the absence of attrition rate reporting and the use of unadjusted reported measures of association. Regardless, we confirmed that all studies are eligible to be included in the review.

## Synthesis

While our initial aim was to perform meta-analyses encompassing children up to 5 years, it was observed that nearly all the included studies assessed growth outcomes in children aged less than 32 months. The exception was a single study, Bennett et al. (2016), which included four sub-studies and reported the outcomes at the age of five years [58]. In cases where outcomes were repeatedly measured in a longitudinal cohort study, the associations measured at different time points were inherently correlated since they were measured in the same individual. Consequently, including all of these estimates in the meta-analysis would have artificially reduced the CIs [59]. To address this, we selected one of the estimates in such cases [59]. To ensure homogeneity in the timepoints of outcomes, the primary meta-analysis focused on estimates reported closest to the one-year mark. For example, in Patel et al. (2003), we selected the adjusted OR (aOR) among children aged six months rather than 6 weeks. For estimates repeated at various time points, we conducted the subgroup analyses to understand the trend of the association based on the outcome timepoints.

Due to the limited number of studies using a consistent measure of association for overweight/obesity, the primary meta-analysis focused only on stunting, underweight, and wasting. Additionally, recognizing the heterogeneity, we stratified the analysis based on the type of measure of association (RR and OR). Meta-analyses were conducted using both the DerSimonian and Laird methods (DL) under a random-effects model and the Knapp–Hartung (KH) standard-error adjustment estimates (also known as Sidik–Jonkman adjustment), given the relatively small number of studies for each outcome [60, 61]. We presented the results of KH estimates in the S3 Table. We reported the overall effect of PND on each adverse growth outcome and performed subgroup analyses based on the timing of depression diagnosis (APD vs. PPD), time point of outcome (4 months, 6 months, 1 year, 2 years, and 5 years), region (Africa, America, Asia, and Middle-East), and study design (case-control, cohort, cross-sectional, and secondary analysis in RCT).

Sensitivity analyses were conducted to assess the influence of 1) unadjusted estimates on the pooled estimate, 2) inclusion of maternal depression measured up to 24 months, and 3) studies published more than ten years ago. Funnel plots and Egger's test were conducted to examine potential publication bias due to missing studies [62]. To evaluate heterogeneity, we followed the guidelines recommended by the Cochrane Statistical Methods Group: $I^2$ 30% to 60% moderate heterogeneity, 50% to 90% substantial heterogeneity, and 75% to 100%

indicating considerable heterogeneity [63, 64]. We also conducted the meta-regression analyses using the type of PND, study year (categorical variable), income levels (low income, lower-middle income, upper-middle income), region, country, study design, and age months of outcome measured (continuous variable) with Sidik–Jonkman random-effects method for each association (RRs and ORs) between PND and outcomes (stunting and underweight). All analyses were performed using Stata Version 17 [65].

## Results

### Description of studies

During the initial database search in 2021, we identified 1,944 records and assessed titles and abstracts of 1,483 articles (Fig 1). From this set, we conducted the full-text review of 128 articles, except for four studies we could not locate (e.g. the conference abstract). In cases where specific estimates required for our research question were not reported (missing estimates for the research question), particularly when maternal depression was measured up to 5 years after birth, we contacted the authors to obtain the relevant estimate (n = 24). Among them, one group, Prado et al. (2019), provided the requested estimate (contacted person and date: Prado and September 19, 2021) [66]. Additionally, three studies, Surkan et al. (2008), Kaaya et al. (2016), and Wemakor and Iddrisu (2018), that measured PPD beyond the upper limit of

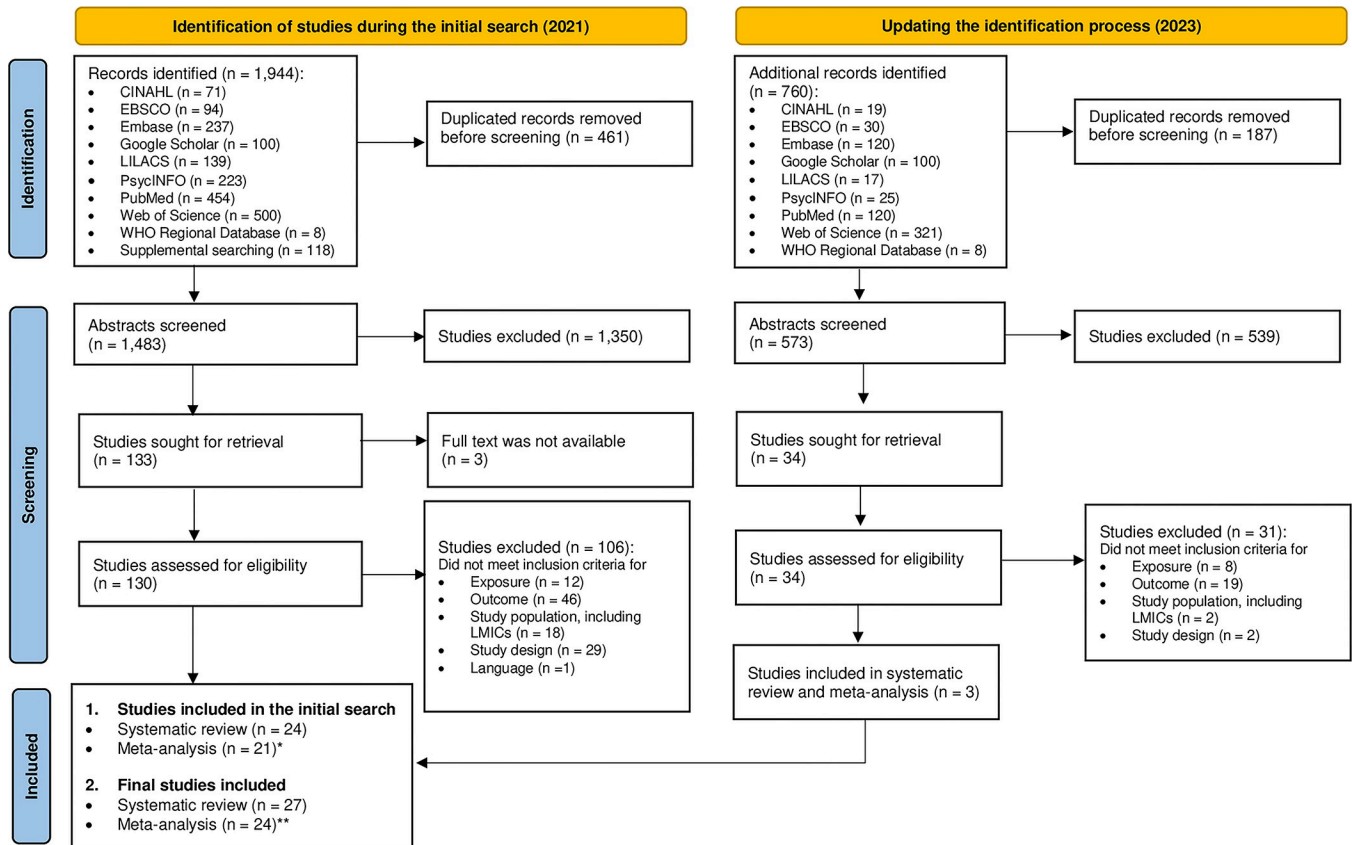

**Fig 1. PRISMA flowchart.** *From*: Page MJ, McKenzie JE, Bossuyt PM, Boutron I, Hoffmann TC, Mulrow CD, et al. The PRISMA 2020 statement: an updated guideline for reporting systematic reviews. BMJ 2021;372:n71. doi: 10.1136/bmj.n71. For more information, visit: http://www.prisma-statement.org/. Note: LMICs = low-income and middle-income countries. *A total of 24 sub-studies included since Bennett et al. (2016) conducted in 4 different countries and reported the findings in each country. *A total of 27 sub-studies included since Bennett et al. (2016) conducted in 4 different countries and reported the findings in each country.

our inclusion criteria (up to 24 months) were included in the systematic review and meta-analysis, and we excluded them in the sensitivity analysis.

Three studies, namely Nasreen et al. (2013), Brentani and Fink. (2016), and Christodoulou (2019), met the other inclusion criteria but did not report the variance of the estimate [34, 56, 67]. Consequently, these studies with missing variance of estimates were included in the systematic review but excluded from the meta-analysis. One study by Bennett et al. (2016), combined sub-studies conducted in India, Ethiopia, Peru, and Vietnam, was analyzed separately [58]. As a result, the systematic review encompassed 24 studies (including 27 sub-studies), while the meta-analysis included 21 studies (comprising 24 sub-studies) during the initial search [34, 35, 56, 58, 66–85].

During the search update in September 2023, we identified 760 studies and screened 573 studies based on their titles and abstracts (Fig 1). Among these, we conducted a full-text review of 34 studies, ultimately including three studies in the systematic review and meta-analysis [39, 86, 87]. As a result, across the two database search processes, a total of 26,261 mother-baby pairs from 27 studies (30 sub-studies) done in 15 countries were included in the final systematic review [34, 35, 39, 56, 58, 66–87] (Table 1). These 27 studies covered 24 studies on stunting, 19 studies on underweight, two studies on overweight/obestiy, and five studies on wasting. In the final meta-analysis, we pooled estimates from a total of 27 sub-studies (n of studies = 24) [35, 58, 66, 68–85]: 21 studies (24 sub-studies) were on stunting, 17 studies (20 sub-studies) on underweight, and three studies on wasting (S2 Data).

Among the total studies included in the systematic review (n = 27), fourteen (52%) were cohort studies, nine (33%) were cross-sectional, three (11%) were case-control studies, and 1 (4%) was a secondary data analysis (cohort study) within an RCT. Only seven studies (26%) utilized APD, including those that measured depression at birth or within two days after birth. All studies that reported the RR between APD and stunting or underweight were cohort studies. Eighteen studies (67%) measured depression as PPD, and two studies (7%) assessed both APD and PPD.

Among the studies included in the meta-analysis for the association between PND and stunting (n = 21), 14 (67%) reported ORs and 7 (33%) studies reported RRs; seven APD studies (33%), 12 PPD studies (57%), one study (5%) that examined APD and PPD separately, and one study (5%) that used PND as a combined exposure. In the case of the meta-analysis for the association between PND and underweight, which included 17 studies, 10 (59%) reported ORs, and 7 (41%) reported RRs. Among these, 3 studies focused on APD (18%), 13 on PPD (76%) studies, and one study (6%) examined APD and PPD separately. For the meta-analysis of the association between PND and wasting, which encompassed 3 studies, all reported RRs: 1 APD study (33%) and 2 PPD (67%) studies.

Various screening tools, cutoff scores, and clinical diagnostic measures were employed to identify mothers with PND. Among the utilized questionnaires, the Edinburgh Postnatal Depression Scale (n = 11) was the most common measure of depression, followed by the Self-Reporting Questionnaire-20 (n = 4), Center for Epidemiological Studies Depression scale (n = 3), Hopkins Symptom Checklist (n = 2), Aga Khan University Anxiety and Depression Scale (n = 1), Depression Anxiety Stress Scales (n = 1), and Pitt Depression Inventory (n = 1) (Table 1). In most cases, the authors described in-detail the reasons for the selected questionnaire, such as reliability and validity, and for the selected cutoff score, the process of the local adaptation, or the use of an interpreter due to literacy issues. The semi-structured or structured clinical interview, such as the Schedules for Clinical Assessment in Neuropsychiatry (n = 1), and Structured Clinical Interview for Diagnostic and Statistical Manual of Mental Disorders-III-R or -IV (n = 3), were also applied by the trained researchers or psychiatrists.

**Table 1. Summary of the study characteristics included in the systematic review and meta-analysis for the association between perinatal depression and child adverse growth outcomes (stunting, underweight, overweight/obesity) in the low-income and middle- income countries (N = 26,261 from 27 studies).**

| First author (year) | Country | Study design | Sample size | Depression measure (cutoff) | Time of depression measure | Association measure | Time of outcome measure (after-birth)[a] | Stunting estimate (95% CI) | Underweight/ overweight or obesity [b] estimate (95% CI) |
|---|---|---|---|---|---|---|---|---|---|
| Patel (2003) | India | Cohort | 171 | EPDS ($\geq$11) | 6 weeks postpartum | aOR | 6 weeks | 1.9 (0.5, 6.8) | 3.2 (1.1, 9.4) |
| | | | | | | | 6 months | 3.2 (1.1, 8.9) | 2.8 (1.1, 7.3) |
| Rahman (2004, 1) | Pakistan | Case-control | 172 | SRQ20 ($\geq$ 10) | > 6 weeks postpartum | aOR | 9 months | | 2.8 (1.2, 6.8) |
| Rahman (2004, 2) | Pakistan | Cohort | 265 | SCAN | 3rd trimester | aOR | 6 months | 3.2 (1.1, 9.9) | 3.5 (1.5, 8.6) |
| | | | | | | | 12 months | 2.8 (1.3, 6.1) | 3.0 (1.5, 6) |
| Anoop (2004) | India | Case-control | 144 | SCID for DSM-III-R | < 6 weeks postpartum | aOR | 6–12 months | | 7.4 (1.6, 38.5) |
| Tomlinson (2006) | South Africa | Cohort | 147 | SCID for DSM-IV | 2 months postpartum | OR | 18 months | 2.52 (0.98, 6.47) | 2.32 (0.90, 6.00) |
| Surkan (2008) | Brazil | Cross-sectional | 595 | CES-D ($\geq$16) | 6–24 months postpartum [c] | aOR | 6–24 months | 1.8 (1.1, 2.9) | 1.8 (0.6, 5.6) |
| Adewuya (2008) | Nigeria | Case-control | 242 | SCID for DSM-III-R | < 6 weeks postpartum | OR | 6 weeks | 1.55 (0.43, 5.65) | 2.60 (0.87, 7.62) |
| | | | | | | | 3 months | 3.28 (1.03, 10.47) | 3.19 (1.21, 8.40) |
| | | | | | | | 6 months | 3.34 (1.18, 9.55) | 4.21 (1.36, 13.20) |
| | | | | | | | 9 months | 2.68 (0.82, 8.80) | 2.84 (0.98, 8.24) |
| Avan (2010) | South Africa | Cohort | 892 | PDI ($\geq$19) | > 6 weeks postpartum | aRR | 24 months | 1.61 (1.02, 2.56) | 0.76 (0.39, 1.49) |
| Ndokera (2010) | Zambia | Cross-sectional | 278 | SRQ20 ($\geq$8) | > 6 weeks postpartum | RR | 2–12 months | 1.48 (0.35, 6.22) | 1.48 (0.35, 6.22) |
| Nasreen (2013)[h] | Bangladesh | Cohort | 652 | EPDS | Third trimester | Adjusted beta-coefficients | 6–8 months | -0.26[i] (-0.50, -0.01) | |
| | | | | | 2–3 months postpartum | | | -0.26[i] (-0.54, 0.01) | -0.38[i] (-0.72, -0.05) |
| Brentani and Fink (2016)[h] | Brazil | Cross-sectional | 798 | EPDS (>13) | 1 year | Adjusted beta-coefficients | 1 year | 0.02 (SE 0.03) | 0.01 (SE 0.05) |
| Kaaya (2016) | Tanzania | Cohort | 699 | HSCL-25 | During pregnancy | aRR | < 2 years | 0.83 (0.67, 1.02) | 0.86 (0.68, 1.09) |
| | | | | | <2 years postpartum [c] | | | 0.86 (0.72, 1.03) | 0.80 (0.54, 0.99) |
| | | | | | Cumulative postpartum | | | 1.03 (1.01, 1.04) | 1.02 (1.00, 1.04) |
| Madeghe (2016) | Kenya | Cross-sectional | 200 | EPDS ($\geq$13) | 6–16 weeks postpartum | aOR | 6–16 weeks | | 5.79 (2.14, 15.62) |
| Upadhyay and Srivastava (2016) | India | Cohort | 1,833 | SRQ20 ($\geq$8) | < 6 weeks postpartum | aOR | 5–21 months | 1.53 (1.21, 1.92) | |

(*Continued*)

**Table 1.** (Continued)

| First author (year) | Country | Study design | Sample size | Depression measure (cutoff) | Time of depression measure | Association measure | Time of outcome measure (after-birth)[a] | Stunting estimate (95% CI) | Underweight/ overweight or obesity[b] estimate (95% CI) |
|---|---|---|---|---|---|---|---|---|---|
| Bennett (2016) | India | Cohort | 1,930 | SRQ20 (≥8) | > 6 weeks postpartum | aRR | 1 year | 1.18 (1.03, 1.35) | 1.11 (0.97, 1.26) |
| | India | | 1,930 | | | | 5 years | 1.07 (0.95, 1.22) | 1.16 (1.00, 1.29) |
| | Peru | | 1,946 | | | | 1 year | 1.06 (0.93, 1.22) | 0.85 (0.60, 1.19) |
| | Peru | | 1,946 | | | | 5 years | 0.96 (0.85, 1.09) | 0.99 (0.67, 1.46) |
| | Vietnam | | 1,961 | | | | 1 year | 0.91 (0.81, 1.02) | 1.29 (1.03, 1.62) |
| | Vietnam | | 1,961 | | | | 5 years | 1.12 (0.95, 1.33) | 1.19 (0.95, 1.48) |
| | Ethiopia | | 1,885 | | | | 1 year | 0.91 (0.81, 1.02) | 1.01 (0.89, 1.15) |
| | Ethiopia | | 1,885 | | | | 5 years | 0.94 (0.82, 1.09) | 0.95 (0.80, 1.12) |
| Saeed (2017) | Pakistan | Cross-sectional | 325 | AKUADS (≥20) | 12 months postpartum | aOR | < 2 years | 3.15 (1.91, 5.18) | 3.26 (1.99, 5.34) |
| Lima (2017) | Brazil | Cohort | 1,124 | CES-D (≥22) | 2nd trimester | aOR | 12–32 months | 1.09 (0.62, 1.89) | 0.84 (0.54, 1.31) |
| Wemakor and Iddrisu (2018) | Ghana | Cross-sectional | 200 | CES-D (≥20) | 6–23 months postpartum [c] | OR | 6–23 months | 1.05 (0.53, 1.73) | |
| Joshi and Raut (2019) | India | Cross-sectional | 300 | EPDS (≥10) | > 6 weeks postpartum | POR | < 1 year | 1.49 (0.83, 2.65) | 1.08 (0.59, 1.98) |
| Prado (2019) | Ghana | Cohort | 1,003 | EPDS[d] | 6 months postpartum | OR[f] | 18 months | 0.73 (0.38, 1.39) | |
| Christodoulou (2019)[h] | South Africa | Cohort | 470 | EPDS (>13) | At birth | OR[g] | 3 months | 1.67 | 0.75 |
| | | | | | | | 6 months | 0.50 | 0.00 |
| | | | | | | | 9 months | 1.50 | 1.00 |
| | | | | | | | 12 months | 1.13 | 1.00 |
| | | | | | | | 24 months | 0.75 | 0.67 |
| Ahmadi Gharaei (2020) | Iran | Cohort | 470 | DASS[e] | During pregnancy | aRR | 6 months | 1.10 (0.67, 1.81) | 1.71 (1.07, 2.09) |
| Anato (2020) | Ethiopia | Cross-sectional | 232 | EPDS (≥13) | 6 weeks postpartum | aOR | 5–10 months | 2.55 (1.24, 5.25) | |
| Tome (2020) | Zimbabwe | 2ndary data analysis (cohort) of RCT | 4,025 | EPDS (≥12 or suicide) | During pregnancy | aOR | 3 months | 0.83 (0.53, 1.30) | |
| | | | | | | aOR | 6 months | 0.82 (0.53, 1.28) | |
| | | | | | | aOR | 12 months | 1.13 (0.84, 1.52) | |
| | | | | | | aOR | 18 months | 1.10 (0.81, 1.49) | |
| Ricci (2023) | South Africa | 2ndary data analysis (cross-sectional) of RCT | 428 | EPDS (>13) | 6–9 months postpartum | aRR | 6–9 months | 1.02 (0.63, 1.66) | 0.81 (0.41, 1.63) |

*(Continued)*

**Table 1.** (Continued)

| First author (year) | Country | Study design | Sample size | Depression measure (cutoff) | Time of depression measure | Association measure | Time of outcome measure (after-birth)[a] | Stunting estimate (95% CI) | Underweight/ overweight or obesity [b] estimate (95% CI) |
|---|---|---|---|---|---|---|---|---|---|
| Shriyan (2023) | India | Cohort | 1,135 | EPDS (>13) | During pregnancy and delivery within 48 h | aOR | 1 year | 1.72 (1.22, 2.43) | |
| Saleh (2023) | Tanzania | Cohort | 1,739 for height 1,525 for weight | HSCL-8 | During pregnancy | aRR | 1 year | 0.92 (0.83, 1.02) | 0.80 (0.56, 1.16) |

a: For the studies that measured the outcomes repeatedly, we used the measure of association that was reported close to year 1 to prevent the correlation of repeated measurements.

b: Studies except for Brentani and Fink (2016) and Lima et al. (2017) used underweight as an outcome.

c: In the case of PPD and cumulative PPD, the timepoint of maternal depression measure was up until 24 months, which is beyond our inclusion criteria for PND.

d: A locally validated adapted version was used.

e: Authors reported the association between depression score in the DASS and the outcome.

f: The research team constructed an odds ratio and 95% CI based on raw data provided by the original study authors.

h: Studies excluded in the meta-analysis due to heterogeneity of measurement and no availability for CIs or SE.

i: Figs 2 and 3 from Nasreen et al. (2013) showed the mean HAZ and WAZ were lower among mothers with APD compared to mothers without APD, meaning that the negative coefficient predicts the increased risk rather than the protective effect.

Note: AKUADS = Aga Khan University Anxiety and Depression Scale, APD = antepartum depression, aOR = adjusted odds ratio, CES-D = Center for Epidemiological Studies-Depression, DASS = Depression Anxiety Stress Scales, EPDS = Edinburgh Postnatal Depression Scale, HSCL-8/-25 = Hopkins Symptom Checklist-8/-25, PDI = Pitt Depression Inventory, POR = prevalence odds ratio, PND = perinatal depression, PPD = postpartum depression, RCT = randomized controlled trial, aRR = adjusted risk ratio, SCID for DSM-III-R, or -IV = Structured Clinical Interview for Diagnostic and Statistical Manual of Mental Disorders-III-R, or -IV, SCAN = Schedules for Clinical Assessment in Neuropsychiatry, SE = standard error, SRQ20 = Self Reporting Questionnaire-20

Among the total of 30 sub-studies included in the systematic review, six were conducted in India, four from South Africa, three each from Pakistan and Brazil, two from Ethiopia, Ghana, and Tanzania, and one each from Bangladesh, Vietnam, Iran, Kenya, Nigeria, Zambia, Zimbabwe, and Peru: Asia 11 studies, Middle East I study, Africa 14 studies, and America 4 studies. A descriptive summary of all studies included in the analysis and adjusted covariates in each study are shown in Tables 1 and 2, respectively.

## Primary outcomes

**Stunting.** *Meta-analysis of ORs between PND and stunting*

Children aged 6 to 32 months whose mothers had PND had significantly higher odds of stunting, OR 1.63 (n = 14, 95% CI 1.32, 2.02, $I^2$ = 56.0%), compared to those of mothers without PND (Fig 2). A sensitivity analysis that excluded studies with unadjusted ORs or PND measured up to 2 years produced a higher pooled estimate: OR 1.81 (n = 9, 95% CI 1.39, 2.36, $I^2$ = 62.3%). Further, when we limited the estimate to the recent publication, the overall effect remained substantial at 1.64 (n = 6, 95% CI 1.22, 2.19, $I^2$ = 69.4%).

In a subgroup analysis based on the type of PND (Fig 2A), APD and PPD were associated with 1.44 times the odds of stunting (n = 4, 95% CI 1.01, 2.06, $I^2$ = 60.5%) and 1.75 times the odds of stunting (n = 10, 95% CI 1.32, 2.31, $I^2$ = 54.5%), respectively. In a subgroup analysis by region (Fig 2B), studies conducted in Asia showed the greatest overall association of OR 1.95 between PND and stunting (95% CI 1.49, 2.56, $I^2$ = 49.1%) than in Africa or America.

**Table 2. Adjusted covariates in each study included in the systematic review and meta-analysis.**

| Measure of Association | First author (year) | Adjusted variables |
|---|---|---|
| Odds ratio | Patel (2003) | Birthweight |
| | Rahman (2004, 1) | Birthweight, number of children, and socioeconomic status |
| | Rahman (2004, 2) | Low birthweight, breastfeeding, > 5 diarrheal episodes/year, parent education, maternal financial empowerment, relative poverty |
| | Anoop (2004) | Maternal intelligence, low birthweight, SES, duration of exclusive breastfeeding, duration of supplementary breastfeeding, immunization status, mother's literacy |
| | Surkan (2008) | Child age, child gender, birthweight, breast-feeding duration, maternal education attainment, sanitation scale score, SES and living conditions scale score, number of children living in household, participation in Family Health Program, maternal self-efficacy score |
| | Madeghe (2016) | Maternal age, marital status, parity, income |
| | Upadhyay and Srivastava (2016) | Pregnancy intention, social supports, birth size, preterm birth, child sex, ever breastfed, serious illness, mother's age at birth of child, maternal education, maternal working status, antenatal care, iron folic acid tablets, tetanus injection, household head's education status, sex of household head, wealth index, drinking water, toilet facility, income socks, religion, caste, place of residence |
| | Saeed (2017) | Maternal age, education, income, occupation, child gender |
| | Lima (2017) | Monthly family income, maternal age, number of children, maternal schooling |
| | Anato (2020) | Wealth index, education, occupation, household food security, maternal workload |
| | Tome (2020) | Cluster, intervention arms, maternal at birth, mother's height, mother's mid upper arm circumference, child sex, single or multiple birth, household wealth index, presence of functioning hand washing station with water and soap, gender of household head, number of children under-fives, household dietary diversity, mother's HIV status at enrolment |
| | Shriyan (2023) | Gestational age at delivery, age of the respondent, education, SES, parity, weight, height, gestational diabetes, birthweight, and smoking status of participants' husband. |
| Risk ratio | Avan (2010) | SES index, maternal age, child gender, preterm delivery |
| | Kaaya (2016) | Maternal perceived low social support, age, marital status, education, financial dependence, anemia, weight, height, HIV stage at enrollment and counselling session attendance, child's gender, birthweight, gestational age, breastfeeding, HIV status, vitamin A, multivitamin and other disease events. |
| | Bennett (2016) | Maternal schooling, sex of child, child age in months, wealth index |
| | Ahmadi Gharaei (2020) | Stunting: multivitamin intake, maternal education, birth height Underweight: infant sex, having tooth at 6 months, birthweight, infection contraction within 6 months |
| | Ricci (2023) | Maternal age |
| | Saleh (2023) | Maternal age, maternal education, household wealth quintile, marital status, maternal employment status, parity, maternal CD4 category, maternal WHO HIV disease stage, taking ART, trial site, child sex, child age, child twin status, and randomized regimen |

*(Continued)*

**Table 2.** (Continued)

| Measure of Association | First author (year) | Adjusted variables |
|---|---|---|
| Beta-coefficient | Nasreen (2013) | Household SES, infant's sex, infant's weight at 2–3 months, infants' temperament (unadaptable) |
| | Brentani and Fink (2016) | Child age, child sex, twin, born small for gestational age, prematurity, maternal age, mother as a primary caregiver, caregiver primary education, caregiver secondary education, caregiver higher education, caregiver married, asset status |

Note: ART = antiretroviral therapy, HIV = human immunodeficiency virus, SES = socioeconomic status

When we stratified the estimates according to the outcome timepoint (S1 Fig), the effect of APD was strongest when measured around 1 year postpartum (S1A Fig). The effect of PPD was most pronounced when measured before 1 year, and for both, the effect on stunting at each time point decreased after one year (S1B Fig). In a subgroup analysis by study design (S2 Fig), cohort studies showed an attenuated effect of PPD, OR of 1.55, on stunting compared to other study designs (n = 4, 95% CI 0.92, 2.63, $I^2$ = 62.4%).

Meta-analysis of RRs between PND and stunting

Children aged 6 to 24 months whose mothers had PND had similar risks of stunting, with an RR 0.99 (n of sub-studies = 10, 95% CI 0.90, 1.08, $I^2$ = 56.0%), compared to children of mothers without PND. In the study by Kaaya et al. (2016), among the three estimates related to the timing of PND (APD, PPD, and cumulative PPD), we selected the one measured for APD that met the inclusion criteria of our study. Sensitivity analyses produced estimates that were similar to those pooled in the main analysis.

The subgroup analyses based on the type of PND, region, timepoint of outcome measurement, and study design (Fig 3, S3 and S4 Figs) consistently showed the null effects on stunting, while the overall effect of APD on stunting at each time point showed a decreasing trend as child ages (S4A Fig, n = 3).

Additional findings on the association between PND and stunting

Three additional studies were included in the systematic review. Nasreen et al. (2013), a cohort study in Bangladesh (Asia), reported that mothers with APD had statistically significantly higher risk of stunting among children aged 6–8 months than mothers without APD ("antepartum depressive symptoms predicted infant's stunting"). Brentani and Fink (2016), a cross-sectional study in Brazil, observed a slight increase in the adjusted probabilities of stunting among one-year-old children from mothers with PPD compared to mothers without PPD; however, this difference was not statistically significant. In line with the subgroup analysis of OR based on the timing of stunting measurement, Christodoulou et al. (2019), a cohort study in South Africa, suggested that mothers with APD, measured at birth, had higher odds of stunting among 1-year-old children (OR 1.13) when compared to mothers without APD which was followed by protective odds of stunting among children aged two years (OR 0.75).

**Underweight.** *Meta-analysis of ORs between PND and underweight*

Children aged 6 to 24 months whose mothers had PND exhibited significantly higher odds of underweight, with an OR of 2.65 (n = 10, 95% CI 1.90, 3.68, $I^2$ = 34.5%), when compared to children of mothers without PND (Fig 4). A sensitivity analysis that 1) excluded studies with unadjusted ORs or PND measured up until 2 years and 2) recent publications yielded higher pooled estimates: 1) OR 3.25 (n = 7, 95% CI 2.41, 4.39, $I^2$ = 0.0%) and 2) OR 3.67 (n = 2, 95% CI 2.33, 5.79, $I^2$ = 2.9%).

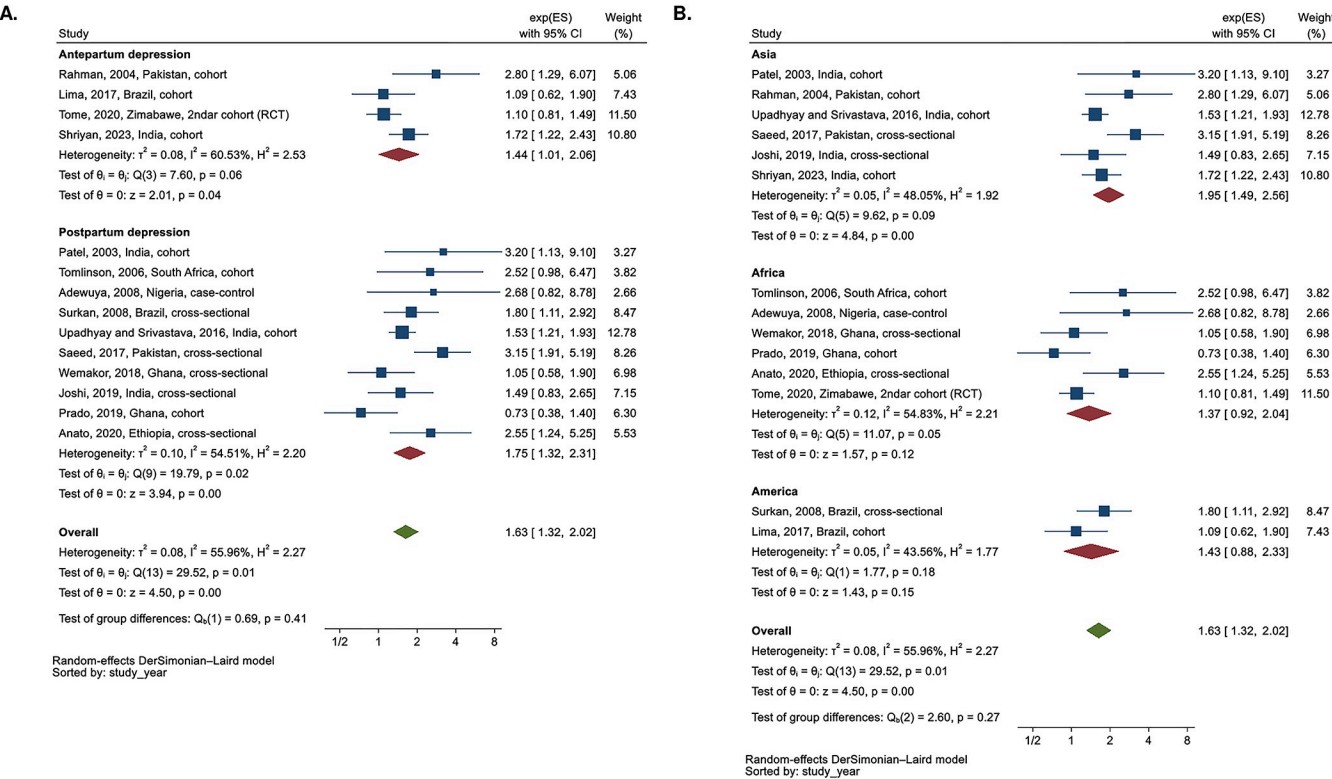

**Fig 2. Meta-analysis of odds ratio between perinatal depression and stunting (n = 14)–subgroup analysis according to type of perinatal depression (antepartum vs. postpartum) and region. A.** Subgroup analysis according to type of perinatal depression. **B.** Subgroup analysis according to region.

The subgroup analysis comparing underweight and APD versus PPD showed a similar trend (Fig 4). In the subgroup analysis based on region (Fig 4B), although the number of included studies was small, the magnitude of the overall OR between PND and underweight followed the order of Africa (n = 3) > Asia (n = 6) > America (n = 1).

When we stratified the estimates according to the timepoint of outcome (S5 Fig), the overall association between PPD and underweight was the highest at the earliest time point (4 months) (n = 1), followed by the decreasing patterns. We presented the subgroup analysis according to the study design in the S6 Fig.

Meta-analysis of RRs between PND and underweight

Children aged 6 to 24 months whose mothers had PND had a similar risk of underweight, with a RR of 1.06 (n of sub-studies = 10 (in Kaaya et al. (2016), we selected APD estimate), 95% CI 0.93, 1.22, $I^2$ = 49.5%), compared to those of mothers without PND. A sensitivity analysis, which excluded studies with unadjusted RRs or PND measured up to 2 years and limited to the recent publications, produced almost the same pooled estimates as the main analysis.

The subgroup analysis comparing underweight and APD versus PPD yielded results that closely resembled the overall analysis (Fig 5A). In the subgroup analysis based on the region (Fig 5B), the effect direction varied by region. In America, one study indicated a potentially protective effect on child underweight among mothers with PND compared to those without PND, though this finding was not statistically significant. Conversely, the overall effect in Asia (RR 1.16 (95% CI 1.01, 1.33)), Africa (RR 1.02 (95% CI 1.00, 1.03)) and one study from the Middle East suggested a higher risk of child underweight among mothers with PND compared to their counterparts.

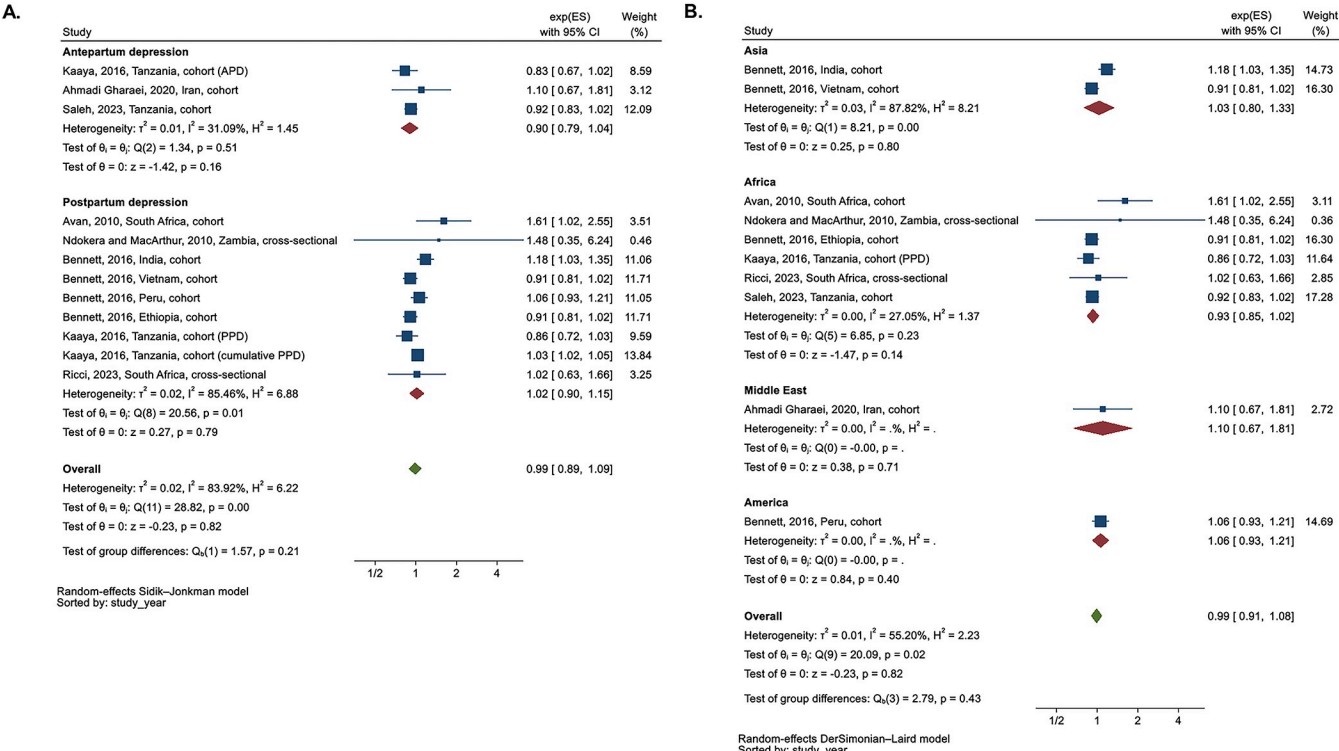

**Fig 3. Meta-analysis of the risk ratio between perinatal depression and stunting of children. A.** Subgroup analysis according to type of perinatal depression [a]. **B.** Subgroup analysis according to region.

We presented the subgroup analysis stratified based on the exposure and outcome time-point and study design in S7 and S8 Figs.

Additional findings on the association between PND and underweight

Three additional studies were included in the systematic review. Nasreen et al. (2013) reported statistically significantly higher risk of underweight among children aged 6–8 months ("Maternal postpartum depressive symptoms independently predicted infant's underweight") (67). Christodoulou et al. (2019) presented percent proportions suggesting a null or protective effect of APD measured at birth on underweight over 2 years following birth.

**Wasting.**   Five studies that met the inclusion criteria reported wasting with heterogeneous measures of association (34,75,77,83,87) (Table 3). Joshi and Raut (2019) in India reported an almost null prevalence OR for wasting among infants aged less than one year when comparing mothers with and without PPD (prevalence OR = 1.01, 95% CI 0.51, 2.01). In South Africa, Christodoulou showed variations in wasting among mothers with and without APD over two years.

In Kaaya et al., only cumulative PND showed a statistically significant association with wasting among mothers with HIV (aRR 1.08 (95% CI 1.04, 1.12)) among three exposures of APD, PPD, and cumulative PPD. A meta-analysis of RRs from Avan et al. (2010), Kaaya et al. (2016), and Saleh et al. (2023) suggested contrasting effects of APD and PPD on the risk of wasting. The RR of APD on wasting was 1.49 (95% CI 0.63, 3.52, $I^2$ = 68.0%), while the RR of PPD on wasting was 0.81 (95% CI 0.51, 1.29, $I^2$ = 69.2%) (Fig 6).

**Overweight/obesity.**   Only two studies reported overweight or obesity as an outcome. Lima et al. (2017), a cohort study in Brazil, found that mothers with and without APD had an OR of 0.84 for overweight among children aged 12 to 32 months (95% CI 0.54, 1.31),

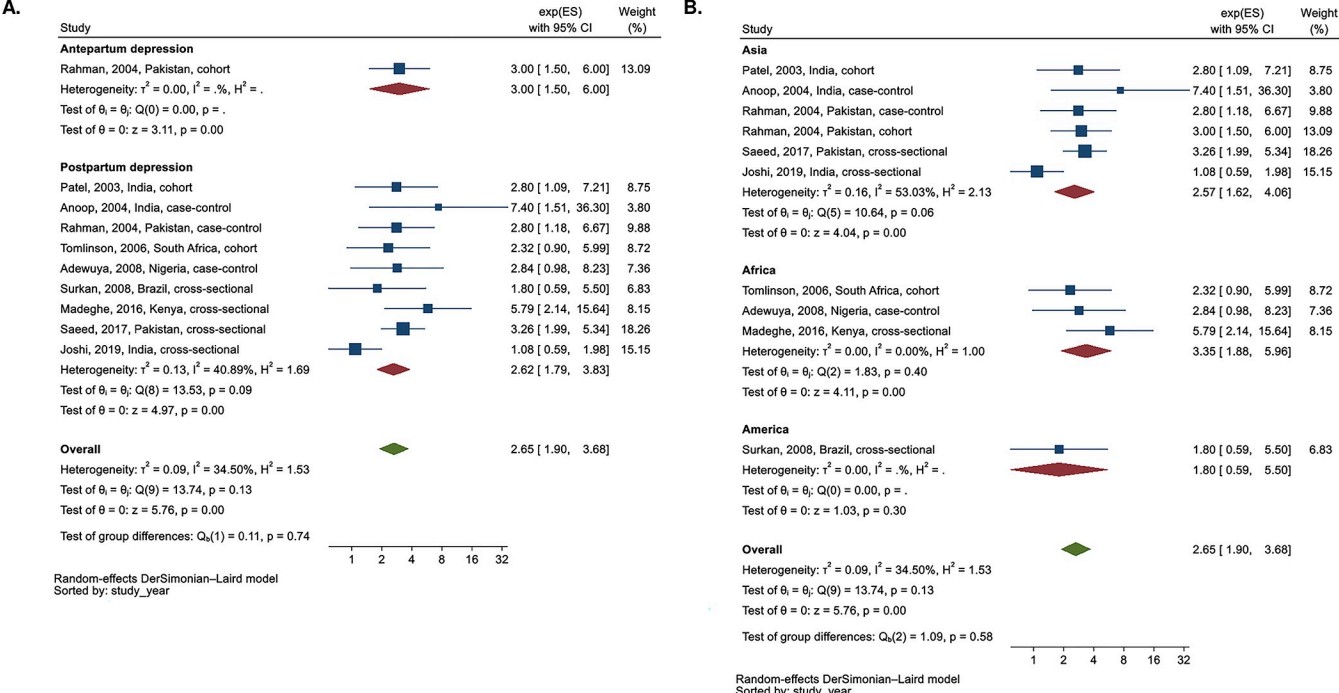

**Fig 4. Meta-analysis of odds ratio between perinatal depression and underweight (n = 10)–subgroup analysis according to region and type of perinatal depression (antepartum vs. postpartum). A.** Subgroup analysis according to type of perinatal depression. **B.** Subgroup analysis according to region.

suggesting a potential protective effect of APD on overweight. On the contrary, Brentani and Fink (2016), a cross-sectional study in Brazil, reported a slight increase in the adjusted probabilities of obesity among children aged 1 year among mothers who had PPD compared to those without PPD, potential risk of PPD on obesity [56]. However, both were not statistically significant.

## Meta-regression

In the meta-regression analyses stratified for association type (RRs and ORs) and outcome types (stunting and underweight) (Table 4), the country explained the largest proportion of between-study variance ($R^2 = 74.7\%$) when examining the OR between PND and stunting. The region of Africa (p-value = 0.046) and the secondary analysis of RCT (p-value = 0.010) were found to be statistically significant predictors for the OR between PND and stunting. For the OR between PND and underweight, the study year explained the highest proportions of the between-study variance ($R^2 = 56.7\%$, p-value for 2019 = 0.058). For the RR between PND and stunting, the study year emerged as a significant predictor ($R^2 = 34.9\%$, p-value for the year 2010 = 0.039), and for the RR between PND and underweight, 1) country, 2) region, and 3) study year played significant roles: 1) $R^2$ of countries = 79.5%, p-value for Iran = 0.003, p-value for Vietnam = 0.025, 2) $R^2$ of region = 49.8%, p-value for Africa = 0.011, and 3) $R^2$ of year = 42.9%, p-value for year 2020 = 0.008).

## Publication bias

According to the measure of association and outcome strata, we obtained p-values in the Egger's tests: OR of stunting (n = 14, p = 0.241), RR of stunting (n of sub-studies = 10, p = 0.292), OR of underweight (n = 10, p = 0.403), and RR of underweight (n of sub-

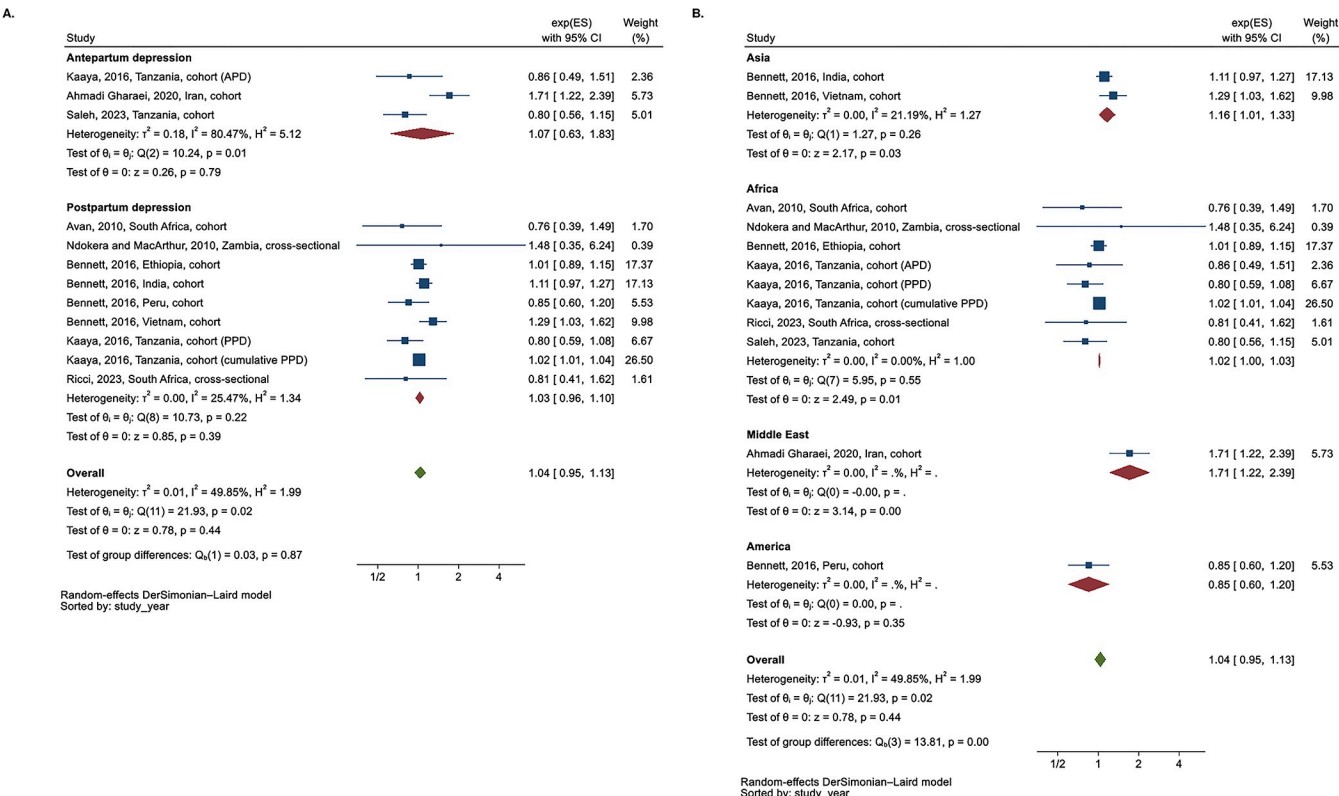

**Fig 5. Meta-analysis of the risk ratio between perinatal depression and underweight of children. A.** Subgroup analysis according to type of perinatal depression [a]. **B.** Subgroup analysis according to region.

studies = 10, p = 0.685). However, visual inspection of the funnel plot reveals substantial asymmetries in the pooled associations found in this study, suggesting the presence of publication bias due to missing studies (S9 Fig).

## Discussion

This systematic review and meta-analysis included 27 studies published from 2003 to 2023 in LMICs. We could not conduct a meta-analysis for obesity/overweight due to the limited numbers of relevant studies, and the association between PND and the rest of adverse growth outcomes exhibited variations based on the measure of association, country/region, PND types, and study design.

Among the studies reporting ORs, we observed that exposure to PND elevated the odds of stunting throughout the initial 32 months of a child's life by 63%, while the influence of PPD alone was even more pronounced, increasing the odds of stunting by 75%. Stunting is unique in its association with fetal development during pregnancy, setting it apart from other forms of undernutrition. This condition, intertwined with both the woman's nutritional status during pregnancy and the baby's nutritional status after birth, can be primarily prevented through optimal maternal health and child feeding and care practices during the first 1,000 days [8]. PND exhibited an even more pronounced effect on children being underweight, with an OR of 2.73 among the studies reported ORs. Being underweight often indicates insufficient nutrient intake and can result from many environmental and genetic factors.

**Table 3. Summary of the study characteristics included in the systematic review and meta-analysis for the association between perinatal depression and wasting of children in low-income and middle-income countries.**

| First author (year) | Country | Study Design | Sample size | Depression measure (cutoff) | Time of depression measure | Association measure | Time of outcome measure (after-birth) | Wasting (95% CI) |
|---|---|---|---|---|---|---|---|---|
| Avan (2010) | South Africa | Cohort | 892 | PDI (≥19) | > 6 weeks postpartum | aRR | 24 months | 0.66 (0.30, 1.42) |
| Kaaya (2016) | Tanzania | Cohort | 699 | HSCL-25 | During pregnancy | aRR | < 2 years | 1.06 (0.74, 1.51) |
|  |  |  |  |  | <2 years postpartum[a] |  |  | 0.59 (0.35, 1.01). |
|  |  |  |  |  | Cumulative postpartum |  |  | 1.08 (1.04, 1.12) |
| Joshi and Raut (2019) | India | Cross-Sectional | 300 | EPDS (≥10) | > 6 weeks postpartum | POR | < 1 year | 1.01 (0.51, 2.01) |
| Christodoulou (2019)[c] | South Africa | Cohort | 470 | EPDS (>13) | At birth | OR[b] | 3 months | 1.45 |
|  |  |  |  |  |  |  | 6 months | 1.50 |
|  |  |  |  |  |  |  | 9 months | 0.00 |
|  |  |  |  |  |  |  | 12 months | 0.50 |
|  |  |  |  |  |  |  | 24 months | 2.00 |
| Saleh (2023) | Tanzania | Cohort | 1,739 for height 1,525 for weight | HSCL-8 | During pregnancy | aRR | 1 year | 2.61 (1.03, 6.65) |

a: In case of PPD and cumulative PPD, the timepoint of maternal depression measure was until 24 months, which is beyond of our inclusion criteria for PND.

b: The authors reported the risk of each stratum of exposure and outcome, and the research team calculated ORs. Since the CIs were not presented, these estimates are only included in the systematic review.

c: Studies excluded in the meta-analysis due to heterogeneity of measurement and no availability for CIs or SE.

Note: CIs = confidence intervals, OR = odds ratio, EPDS = Edinburgh Postnatal Depression Scale, HSCL-25 = Hopkins Symptom Checklist, PDI = Pitt Depression Inventory, POR = prevalence odds ratio, aRR = adjusted risk ratio, SE = standard error

For both stunting and underweight, studies in India and Pakistan consistently reported high associations with depression [68, 70, 71, 80, 88]. However, studies from Iran and Vietnam also found high associations of PND for the risk of underweight and PND but not for the risk of stunting [58, 84]. While the pooled RR analyses of stunting and underweight differed from the pooled OR analyses, meta-regression showed factors like region, country, and study year explained these results. India and Pakistan have historically had the highest rates of stunting (more than 30%) among children for decades, and women in these regions often experience excessive workloads and high stress levels, particularly during perinatal periods [15, 89]. In line with this, Holm-Larsen et al. (2019), a cohort study in Tanzania, where the prevalence of stunting was high (35% during the study period) (S1 Table) [15], reported that the mean HAZ of 2 to 3-year-old children whose mothers had PPD was significantly lower by 0.32 units (95% CI -0.49, -0.15) compared to children whose mothers did not have PPD [28]. However, in Brazil, where the prevalence of stunting was much lower at 3.6%, Santos et al. (2010) reported no association between maternal PPD symptoms and stunting among 4-year-old children (S1 Table) [44].

Additionally, we could not obtain a consistent association between wasting and PND. Wasting, which heightens the risk of death in young children due to recent, severe weight loss, is a relatively rare outcome with a global prevalence of 6.7% compared to stunting at 22% among children under the age of five [15, 90]. The two studies meeting our criteria resulted in inconclusive associations, even with the qualitative analysis [75, 83]. Results from the studies

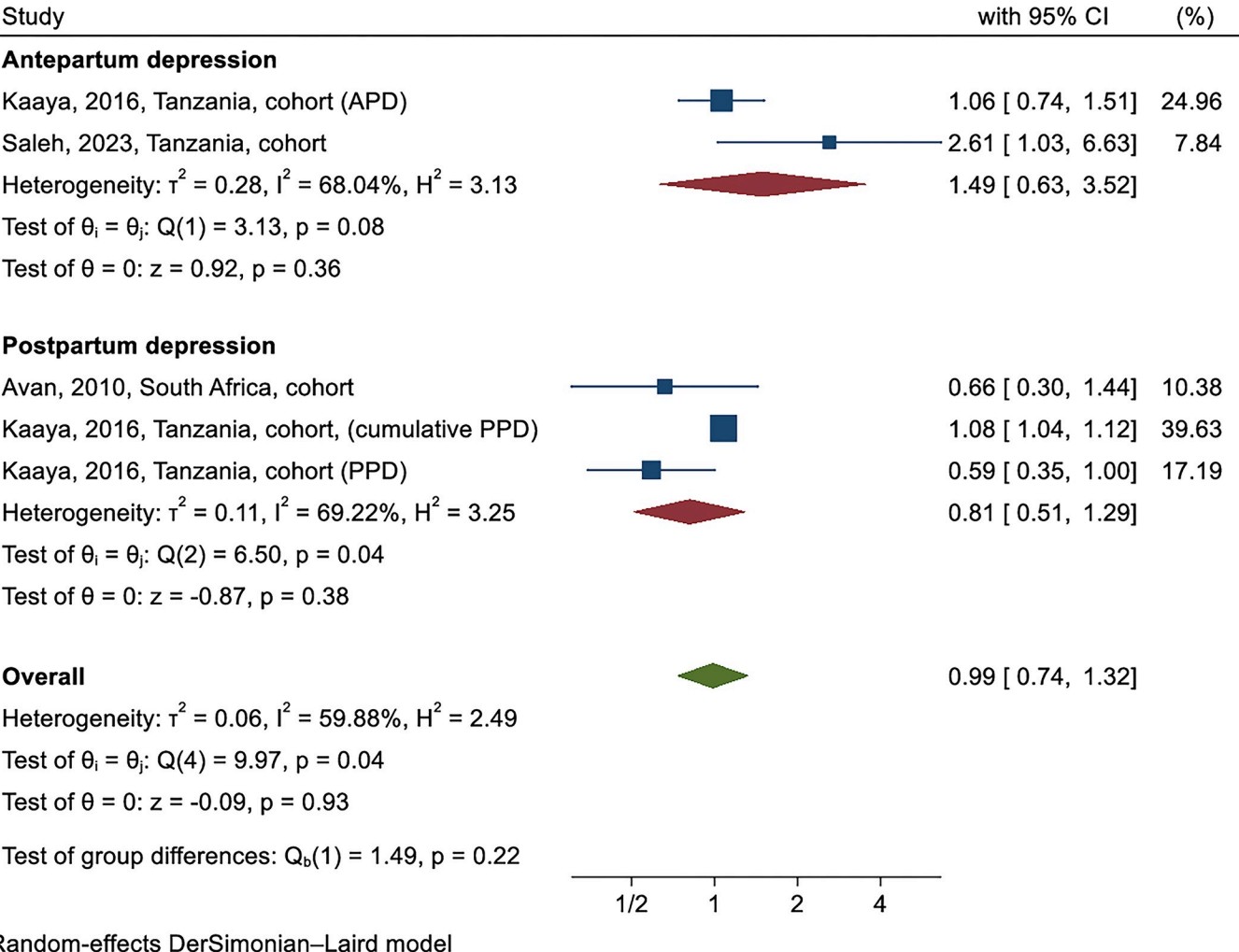

**Fig 6. Subgroup analysis of the risk ratio between perinatal depression and wasting according to type of perinatal depression [a].**

excluded due to the heterogeneity of measures of association also presented conflicting findings. For example, in a cohort study in Tanzania (N = 1,128), the adjusted mean z-score of weight for height was significantly 0.21 units higher (95% CI 0.02, 0.40) among 2 to 3-year-old children with mothers with PPD compared to children without PPD, suggesting that PPD may be protective against wasting [28]. A Brazilian cohort study (N = 3,792), which we excluded due to the timing of outcome measurement, reported heterogeneous association according to PPD duration (12, 24, and 48 months) [44]. Given that wasting is an acute outcome of malnutrition, wasting may be more influenced by acute food crises, such as natural and man-made disasters, rather than maternal depression-related dysfunction. Additionally, in some LMICs, where the traditional extended family system remains prevalent, the risk of children experiencing wasting might be mitigated by the support of extended family members [89].

In light of the growing worldwide obesity epidemic, there is emerging evidence that PND may be associated with childhood overweight and obesity, although research in LMICs remains limited [91–93]. The mechanisms of child overweight and obesity are not fully

**Table 4. Table of estimates from the meta-regression.**

| Association | Variable | Test of residual homogeneity | | I² (%) | R² (%) |
|---|---|---|---|---|---|
| | | Q residual | p-value | | |
| Log OR between PND and stunting | APD vs. PPD | 40.65 | 0.004 | 66.8 | 11.2 |
| | Study year | 21.15 | 0.048 | 60.2 | 24.6 |
| | Income level | 49.27 | 0.000 | 71.4 | 0.0 |
| | Region | 35.46 | 0.012 | 66.4 | 10.9 |
| | Country | 8.05 | 0.887 | 37.9 | 74.7 |
| | Study design | 26.53 | 0.088 | 60.8 | 31.5 |
| | Age months of outcome measured | 50.52 | 0.000 | 71.5 | 0.0 |
| Log OR between PND and underweight | APD vs. PPD | 14.63 | 0.331 | 31.5 | 0.0 |
| | Study year | 2.43 | 0.965 | 12.8 | 56.7 |
| | Income level | 14.38 | 0.347 | 31.0 | 0.7 |
| | Region | 13.54 | 0.331 | 32.2 | 0.0 |
| | Country | 8.25 | 0.509 | 31.17 | 0.0 |
| | Study design | 13.38 | 0.342 | 30.6 | 0.0 |
| | Age months of outcome measured | 14.84 | 0.318 | 30.3 | 0.0 |
| Log RR between PND and stunting | APD vs. PPD | 25.95 | 0.026 | 78.6 | 3.7 |
| | Study year | 24.76 | 0.016 | 74.1 | 34.9 |
| | Income level | 27.57 | 0.010 | 78.1 | 0.0 |
| | Region | 32.16 | 0.001 | 78.7 | 0.0 |
| | Country | 20.30 | 0.009 | 73.2 | 20.1 |
| | Study design | 32.79 | 0.003 | 82.4 | 0.0 |
| | Age months of outcome measured | 32.63 | 0.003 | 81.5 | 0.0 |
| Log RR between PND and underweight | APD vs. PPD | 29.43 | 0.009 | 83.7 | 0.0 |
| | Study year | 18.79 | 0.093 | 76.1 | 42.9 |
| | Income level | 28.15 | 0.009 | 79.5 | 5.8 |
| | Region | 8.27 | 0.764 | 70.7 | 49.8 |
| | Country | 5.71 | 0.680 | 34.6 | 79.5 |
| | Study design | 30.20 | 0.007 | 84.2 | 0.0 |
| | Age months of outcome measured | 28.82 | 0.011 | 81.7 | 0.0 |

Note. APD = antepartum depression, OR = odds ratio, PND = perinatal depression, PPD = postpartum depression, RR = risk ratio

understood but are believed to include a complex interplay of genetic, lifestyle, and environmental factors [94]. In LMICs, which face the challenge of a dual burden of malnutrition, overweight was initially associated with higher SES segments of society, while underweight was more common among the poor. However, as energy-dense, nutrient-poor diets become increasingly prevalent among lower SES groups, the prevalence of overweight has also risen [95]. We observed that only one study on overweight/obesity met the criteria for analysis, making it difficult to discern clear patterns. In addition to Lima et al. (2017), Santos et al. (2010) reported no association between PPD and overweight among 4-year-old children whose mothers had PPD symptoms once or twice over during the measurement period [44]. However, when mothers experienced PPD consistently, four-year-old children whose mothers had PPD had 1.6 times the adjusted odds of being overweight (95% CI 1.0, 2.5) compared to children who had mothers without PPD. Consequently, further research is needed to investigate the relationship between PND and overweight/obesity among children in LMICs.

Most meta-analyses demonstrated substantial heterogeneity. We observed significant variations in the characteristics of existing studies, including different measures and their

thresholds, types of measure of association, features of the study population (country, region, time), and analytic approaches (selection of confounding factors) (Table 2). The included studies used different tools to measure PND, with different sensitivities and specificities, and this can be one plausible explanation for the observed heterogeneity. Moreover, more than half of the studies incorporated birthweight or preterm birth as adjusting covariates (Table 2). Birthweight or preterm birth, which could be on the causal pathways of PND and outcomes, may inadvertently block one of the causal pathways when we adjust for these factors during the analysis (S10 Fig). Lastly, it is essential to consider that there are often variable lags between depression onset and its clinical diagnosis. A postpartum diagnosis, for instance, does not conclusively rule out the presence of depression during or prior to pregnancy. Untreated APD is a strong predictor of PPD [96]. Roughly one-third of PPD cases have onset during pregnancy, and 27% of PPD started before pregnancy [97]. Therefore, conducting further longitudinal cohort studies that can account for these time-varying characteristics of exposure, outcome, and confounding factors is imperative.

Regardless, among current studies, there is a notable gap in the existing research pertains to assessing the degree of support system, family structure, the extent of a mother's influence on child growth, and food insecurity. The unique characteristics and dynamics of LMICs, shaped by their demographic transitions, imply that the role of the community and support systems in childcare can vary significantly. Children whose mothers experience PND and the challenges of caregiving might have gained more support from the community, like extended families, to compensate for the insufficient support from mothers. Therefore, the context of social support and the meaning of food and economic insecurity could vary according to diverse LMICs, and future studies need to integrate more diverse socio-cultural contexts when investigating this research question [98].

## Conclusion

Limited studies investigated the association between PND and adverse child growth outcomes in LMICs. Nonetheless, mental health is underprioritized and underfunded, especially in most LMICs. Up to 85% of people with severe mental disorders in LMICs receive no treatment, while average per capita spending on mental health is just US $0.25 [99]. These underscore the long-term implications for child health and development and emphasize the disproportionate impact on children in resource-constrained settings. Despite the availability of evidence for supporting low-resource intervention programs, such as the WHO-recommended Thinking Healthy Program in LMICs, there remains a substantial gap in their implementation and scale-up [100]. As eloquently articulated by Howard et al. (2014), 'There is no health without perinatal mental health' [101]. Public health practitioners must prioritize the prevention, screening, and treatment of both prenatal and postpartum depression as integral components of efforts to combat child malnutrition in LMICs within the framework of broader maternal and child health programs.

## Supporting information

**S1 Text. The structured search strategy utilized.**
(DOCX)

**S1 Table. Summary of the findings on the association between maternal depression and child growth outcome.**
(XLSX)

**S2 Table. Quality assessment of studies with the Newcastle Ottawa Scale.**
(XLSX)

**S3 Table. Pooled associations measured using the Knapp–Hartung (KH) standard-error adjustments.**
(XLSX)

**S1 Data. Screened studies.**
(XLSX)

**S2 Data. Data for meta-analysis.**
(CSV)

**S1 Fig. Subgroup analysis of the odds ratio between perinatal depression and stunting according to the time point of stunting measurement. A.** The odds ratio between antepartum depression and stunting according to the timepoint of outcome measurement. **B.** The odds ratio between postpartum depression and stunting according to the timepoint of outcome measurement.
(TIF)

**S2 Fig. Subgroup analysis of the odds ratio between perinatal depression and stunting according to study design. A.** The odds ratio between antepartum depression and stunting according to the study design (n = 4). **B.** The odds ratio between postpartum depression and stunting (n = 10).
(TIF)

**S3 Fig. Subgroup analysis of the risk ratio between postpartum depression and stunting according to study design (n of sub-studies = 8).**
(TIF)

**S4 Fig. Subgroup analysis of the risk ratio between perinatal depression and stunting according to the time point of stunting measurement. A.** The risk ratio between antepartum depression and stunting timepoint of outcome measurement. **B.** The risk ratio between postpartum depression and stunting according to the timepoint of outcome measurement.
(TIF)

**S5 Fig. Subgroup analysis of the odds ratio between postpartum depression and underweight according to underweight timepoint (n = 9).**
(TIF)

**S6 Fig. Subgroup analysis of the odds ratio between postpartum depression and underweight according to study design (n = 9).**
(TIF)

**S7 Fig. Subgroup analysis of the risk ratio between perinatal depression and underweight according to outcome timepoint. A.** The risk ratio between antepartum depression and underweight according to outcome timepoint. **B.** The risk ratio between postpartum depression and underweight according to outcome timepoint.
(TIF)

**S8 Fig. Subgroup analysis of the risk ratio between postpartum depression and underweight according to study design (n of sub-studies = 8).**
(TIF)

**S9 Fig. Contour-enhanced Funnel plots of included studies in the meta-analysis. A.** A funnel plot of studies reporting odds ratios between perinatal depression and stunting. **B.** A funnel plot of studies reporting odds ratios between perinatal depression and underweight. **C.** A funnel plot of studies reporting risk ratios between perinatal depression and stunting. **D.** A funnel plot of studies reporting risk ratios between perinatal depression and underweight.
(TIF)

**S10 Fig. Directed acyclic graph of time-varying characteristics of exposure, mediator-confounders, and outcomes.** Note: APD = antepartum depression, PPD = postpartum depression, L0 = confounder before pregnancy, L1 = time-varying mediator-confounder after delivery (ie, family support). Time-varying exposure: APD and PPD. Time-varying outcome: birth weight and child growth outcome at year 1. Green arrows indicate the causal pathways we want to measure. Red arrows indicate the backdoor path that induces bias when we measure the association between exposure and outcome.
(TIF)

## Acknowledgments

The authors thank Dr. Paul Bain, a librarian of the Harvard Countway Library, for his contributions to the search strategy. We are also grateful to Dr. Catie Oldenburg, the professor of the Department of Epidemiology & Biostatistics at the University of California San Francisco, for her advice on the revision of this manuscript.

## Author Contributions

**Conceptualization:** Elizabeth Carosella, Shradha Chhabria, Hyelee Kim, Aliya Moreira, Dana Naamani, Brennan Ninesling, Bizu Gelaye, Aisha Yousafzai, Stefania Papatheodorou.

**Data curation:** Elizabeth Carosella, Shradha Chhabria, Hyelee Kim, Aliya Moreira, Dana Naamani, Brennan Ninesling, Bizu Gelaye.

**Formal analysis:** Elizabeth Carosella, Shradha Chhabria, Hyelee Kim, Aliya Moreira, Dana Naamani, Brennan Ninesling, Aimee Lansdale, Lakshmi Gopalakrishnan.

**Investigation:** Elizabeth Carosella, Shradha Chhabria, Hyelee Kim, Aliya Moreira, Dana Naamani, Brennan Ninesling, Aimee Lansdale, Lakshmi Gopalakrishnan.

**Methodology:** Shradha Chhabria, Hyelee Kim, Aliya Moreira, Dana Naamani, Brennan Ninesling, Stefania Papatheodorou.

**Supervision:** Bizu Gelaye, Aisha Yousafzai, Stefania Papatheodorou.

**Validation:** Bizu Gelaye, Aisha Yousafzai, Stefania Papatheodorou.

**Visualization:** Elizabeth Carosella, Hyelee Kim, Aliya Moreira, Aimee Lansdale.

**Writing – original draft:** Elizabeth Carosella, Shradha Chhabria, Hyelee Kim, Aliya Moreira, Dana Naamani, Brennan Ninesling, Bizu Gelaye, Aisha Yousafzai, Stefania Papatheodorou.

**Writing – review & editing:** Elizabeth Carosella, Shradha Chhabria, Hyelee Kim, Aliya Moreira, Dana Naamani, Brennan Ninesling, Aimee Lansdale, Lakshmi Gopalakrishnan, Bizu Gelaye, Aisha Yousafzai, Stefania Papatheodorou.

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
