## [Decision Letter · Decision Letter 0]

23 May 2024

PGPH-D-23-02443

Perinatal depression and adverse child growth outcomes in low- and middle-income countries (LMICs): a systematic review and meta-analysis

Dear Dr. Hyelee Kim,

Thank you for submitting your manuscript to PLOS Global Public Health. After careful consideration, we feel that it has merit but does not fully meet PLOS Global Public Health’s publication criteria as it currently stands. Therefore, we invite you to submit a revised version of the manuscript that addresses the points raised during the review process.

We look forward to receiving your revised manuscript.

Kind regards,

Jayanta Kumar Bora, PhD

Academic Editor

Journal Requirements:

Additional Editor Comments (if provided):

Reviewers' comments:

Reviewer's Responses to Questions

**Comments to the Author**

1. Does this manuscript meet PLOS Global Public Health’s publication criteria? Is the manuscript technically sound, and do the data support the conclusions? The manuscript must describe methodologically and ethically rigorous research with conclusions that are appropriately drawn based on the data presented.

Reviewer #1: Yes

Reviewer #2: Yes

Reviewer #3: Yes

2. Has the statistical analysis been performed appropriately and rigorously?

Reviewer #1: Yes

Reviewer #2: Yes

Reviewer #3: Yes

3. Have the authors made all data underlying the findings in their manuscript fully available (please refer to the Data Availability Statement at the start of the manuscript PDF file)?

Reviewer #1: Yes

Reviewer #2: Yes

Reviewer #3: Yes

4. Is the manuscript presented in an intelligible fashion and written in standard English?

Reviewer #1: Yes

Reviewer #2: Yes

Reviewer #3: Yes

5. Review Comments to the Author

Reviewer #1: This was a robust systematic review with good flow of read. I want to commend the authors for the work done. Below are some observations made during the review.

Introduction

Page 4. Paragragh 2 line 2….check the sentence “ there is relatively less evidence exists”. Paragragh 3, line 5 also check the sentence “our primary interest lay understanding the temporal relationship between PND and adverse child growth outcomes”.

Page 4, line 6 what do you mean by occurring before as used in the sentence

Methods

I find some conflicting information in the method and discussion. In the method section, I read articles included where from 2021 to 2023. But the in discussion, you mentioned that the search spanned from 2003 and 2023. Kindly reconcile. Furthermore, considering different tools were used in measuring depression, could it not have affected the results? I mean, they have different validity, strength of measurement and scoring. If so, it can be highlighted as a limitation of study. Some studies on PND usually include data on relationship of the PND and patient or demographic factors. Were these factored into your study?

Discussion

Page 36, paragragh 1, line 10. Check the sentence and rephrase” In the Brazilian cohort study (N = 3,564), which we excluded due to timing of outcome measurement, the authors reported that mothers who scored as having PPD three times (12, 24, and 48 months).”

Page 37, line 11…check sentence “Similar to the issue of with wasting”. Also check line 13 -16.

Line 24…prioritized

Page 38 line 12, we sought,Line 20, 27 % started

Figures

In the figures, I tried understanding the columns exp (ES) and % weight. There are no keys explaining them

Reviewer #2: their investigation entailed meticulous searches throughout a comprehensive array of nine databases, including PubMed, EMBASE, Web of Science, CINAHL Plus, Global Health Database, Google Scholar, WHO Regional Databases, PsycINFO, and LILACS, spanning the period from January 2000 to September 2023. they placed a deliberate focus on studies that utilized well-established screening instruments or clinical interviews to gauge perinatal depression (PND) over the course of pregnancy or up to a year after childbirth. Inclusion criteria were centered on research that detailed any of four detrimental growth indicators in children under five years old, namely stunting, wasting, underweight, and overnutrition/obesity. To synthesize the findings,they integrated both risk ratios (RRs) and odds ratios (ORs) that link PND to these negative growth outcomes, employing a random-effects model to accommodate the diversity inherent in the data.

The paper is very good but very long. Researchers need to focus on the important results, narrow down the details, and transfer some tables to the appendices to make it easier for readers.

Reviewer #3: Perinatal depression and adverse child growth outcomes are major public health concerns in many in low- and middle-income countries (LMICs). Thus, the paper is critical and it contributes to a major research and health contribution. The analyses carried out by the authors are rigorous and merit publication for the knowledge to share with the academic and health community. However, there are major revisions to be made to make the paper very firmed and readable. I have made comments in the paper which could guide authors to effect the changes.

6. PLOS authors have the option to publish the peer review history of their article (what does this mean?). If published, this will include your full peer review and any attached files.

**Do you want your identity to be public for this peer review?** For information about this choice, including consent withdrawal, please see our Privacy Policy.

Reviewer #1: No

Reviewer #2: No

Reviewer #3: **Yes: **David Atombire Adumbire

---

## [Decision Letter · Decision Letter 1]

24 Sep 2024

Perinatal depression and adverse child growth outcomes in low-income and middle-income countries (LMICs): a systematic review and meta-analysis

PGPH-D-23-02443R1

Dear Hyelee Kim,

We are pleased to inform you that your manuscript 'Perinatal depression and adverse child growth outcomes in low-income and middle-income countries (LMICs): a systematic review and meta-analysis' has been provisionally accepted for publication in PLOS Global Public Health. You have to address the points raised by the reviewers before final acceptance.

Best regards,

Jayanta Kumar Bora,PhD

Academic Editor

Reviewer Comments (if any, and for reference):

Reviewer's Responses to Questions

**Comments to the Author**

1. If the authors have adequately addressed your comments raised in a previous round of review and you feel that this manuscript is now acceptable for publication, you may indicate that here to bypass the “Comments to the Author” section, enter your conflict of interest statement in the “Confidential to Editor” section, and submit your "Accept" recommendation.

Reviewer #1: All comments have been addressed

Reviewer #3: All comments have been addressed

2. Does this manuscript meet PLOS Global Public Health’s publication criteria? Is the manuscript technically sound, and do the data support the conclusions? The manuscript must describe methodologically and ethically rigorous research with conclusions that are appropriately drawn based on the data presented.

Reviewer #1: Yes

Reviewer #3: Yes

3. Has the statistical analysis been performed appropriately and rigorously?

Reviewer #1: Yes

Reviewer #3: Yes

4. Have the authors made all data underlying the findings in their manuscript fully available (please refer to the Data Availability Statement at the start of the manuscript PDF file)?

Reviewer #1: Yes

Reviewer #3: Yes

5. Is the manuscript presented in an intelligible fashion and written in standard English?

Reviewer #1: Yes

Reviewer #3: Yes

6. Review Comments to the Author

Reviewer #1: The authors have addressed the comments I raised. They have made adequate adjustments where necessary.

Reviewer #3: Based on my thorough review of the revised submitted manuscript, I have come to the following conclusion and decision.

My general comments are that; 1. the paper is a good piece, 2. the analyses carried out by the authors are rigorous and merit publication for the knowledge to share with the academic and health community and 3. the authors have addressed over 95% of my previous comments made in the earlier draft. While I recommend the paper for onward publication, I suggest that the authors work on redesigning the PRISMA flowchart to make it more refined. They must also be clear on the total number of studies that were included in the whole analysis. See the authors trying to say the total number =27 studies (systematic review) then again will see 24 studies for meta-analysis (n=24→total 27 substudies). This is confusing. Aside this issues, I have no issue with the paper accepted for publication.

I, therefore, recommend that the paper be accepted for publication

7. PLOS authors have the option to publish the peer review history of their article (what does this mean?). If published, this will include your full peer review and any attached files.

**Do you want your identity to be public for this peer review?** For information about this choice, including consent withdrawal, please see our Privacy Policy.

Reviewer #1: **Yes: **Isabel Naomi Aika

Reviewer #3: No
